# Theoretical Study of Thermoelectric Properties of a Single Molecule of Diphenyl-Ether

**Rafael G. Toscano-Negrette** [1,2,*], **José C. León-González** [1,2], **Juan A. Vinasco** [1], **Judith Helena Ojeda Silva** [3,4], **Alvaro L. Morales** [1] and **Carlos A. Duque** [1]

1. Grupo de Materia Condensada-UdeA, Instituto de Física, Facultad de Ciencias Exactas y Naturales, Universidad de Antioquia UdeA, Calle 70 No. 52-21, Medellín 050010, Colombia; jose.leong@udea.edu.co (J.C.L.-G.); juan.vinascos@udea.edu.co (J.A.V.); alvaro.morales@udea.edu.co (A.L.M.); carlos.duque1@udea.edu.co (C.A.D.)
2. Departamento de Física y Electrónica, Universidad de Córdoba, Carrera 6 No. 77-305, Montería 230002, Colombia
3. Grupo de Física de Materiales, Universidad Pedagógica y Tecnológica de Colombia, Tunja 150003, Colombia; judith.ojeda@uptc.edu.co
4. Laboratorio de Química Teórica y Computacional, Grupo de Investigación Química-Física Molecular y Modelamiento Computacional (QUIMOL), Facultad de Ciencias, Universidad Pedagógica y Tecnológica de Colombia, Tunja 150003, Colombia
* Correspondence: rafael.toscano@udea.edu.co

**Abstract:** Taking into consideration the research that has been conducted on the optical and electrical properties of molecular systems, especially the good thermoelectric energy conversion at a nanometric scale that such systems have presented, here we present a new alternative by using a particular diphenyl-ether molecule as a functional device. Such a molecular system is modeled as a planar segment coupled to two electrodes in the first-neighbor approximation within a tight-binding Hamiltonian. We study the electrical and thermal properties of diphenyl-ether molecules such as the electric current, electrical and thermal conductance, Seebeck coefficient, and figure of merit, in the strong and weak coupling regimes, considering different structural configurations and variations with temperature. Our results could be valuable for laboratory applications and/or verification since we characterize the diphenyl-ether molecule as a semiconductor device for different structural models.

**Keywords:** diphenyl-ether; tight-binding; Seebeck coefficient; figure of merit $ZT$

## 1. Introduction

The foundations of molecular electronics began establishment in the 1970s when Aviram and Ratner prepared and characterized a molecular system with a donor–acceptor species by performing electron transfer tests, in order to find devices with rectifying properties when the system was submitted to a potential difference [1]. In the last decades, the interest in the study of low dimensional molecular systems has increased in search of their possible use and application. In particular, systems such as organic molecules have been analyzed by coupling them to electrodes, obtaining conductive, semiconducting, and/or insulating behaviors, making them worthy of being considered for electronic connectors, rectifiers, amplifiers, and/or storage devices [2–5].

Molecular systems have sparked great interest in the field of optoelectronic devices due to their unique properties when interacting with light. These systems, which can consist of individual molecules or molecular assemblies, exhibit exceptional capabilities to absorb, emit, or scatter photons. This light-interacting characteristic of molecular systems has driven their application in the field of renewable energy, particularly in the development of solar cells. Solar cells based on molecular systems offer a sustainable solution to the growing energy demand by harnessing solar light, an unlimited and environmentally friendly energy source. The efficient conversion of sunlight and waste heat into usable

electricity presents a fundamental challenge for researchers in this field. The goal is to fully replace the use of fossil fuels, thereby avoiding environmental pollution and promoting a cleaner and more sustainable means of energy generation [6–9].

Molecular systems can exhibit nonlinear optical properties, which can be utilized in applications such as optical information processing, integrated optics, and telecommunications. Compounds with large delocalized electron systems show exceptionally large nonlinear responses and higher laser damage thresholds compared to inorganic materials. Furthermore, these properties can be modified to optimize additional properties, such as mechanical and thermal stability. An example of such molecular compounds with a significantly large first-order nonlinear optical response is diphenyl-ether [10].

According to the above, in the literature, we find very interesting analyses of thermoelectric and magnetic properties in molecular systems, such as DNA chains, benzene molecules, biphenyl molecules [11–14], and especially theoretical-experimental studies of these properties through the molecular system diphenyl-ether. Motivated by the particular characteristics of this last diphenyl-ether molecular system, especially those reported by Dadosh et al. (2005), where they characterized it as a conducting molecular device, we analyze its thermoelectric properties, taking into account different structural configurations.

To complement the above, it is important to emphasize that Dadosh et al. studied a system of three short organic molecules: 4,4-biphenyldithiol (BPD) conjugated molecule; a bis-(4-mercaptophenyl)-ether (BPE) molecule, in which the conjugate is broken in the center by an oxygen atom; and the 1,4-benzenedimethanethiol (BDMT) molecule, in which the conjugate is broken near the contacts by a methylene group, concluding that the oxygen in BPE and the methylene groups in BDMT suppress electrical conduction relative to BPD [15]. Now, to make our study of the diphenyl-ether properties even more interesting, we highlight the work by S. K. Maiti, who performed a theoretical study of the electron transport properties through single conjugated molecules (BPD, BPE, and BDMT) sandwiched between two non-superconducting electrodes, using the Green's function technique, within a tight-binding model, finding that the electron transport properties are significantly influenced by the existence of localized groups in these conjugated molecules and the coupling strength between the molecule and the electrode [16].

On the other hand, we know that to analyze any system of low dimensionality—leading to a characterization through electrical, thermal, spintronic, structural properties, among others—there are many methods used both theoretically and experimentally. For example, at the experimental level, we find the mechanically controlled breakage joints (MCBJ), which search for the relationship of the electrical conductance as a function of the electrode separation, the scanning tunneling microscope (STM), which seeks to characterize the tunneling current with the application of a voltage [17–25], among others. On the side of theoretical methods, we find, among many, two important ways: one (strictly numerical) that includes the application of quantum correlation functional theory through effective equations of a single particle, also known as the density functional theory (DFT), and another (numerical and/or analytical) that includes a process of renormalization of the real space by using Green's functions, where the degrees of freedom of the system are reduced to obtain a one-dimensional system that contains all the structural information of the quantum system [26–30].

It is important to remark that with the two theoretical methods mentioned above, it is possible to determine both the electrical and thermal properties, as well as the spintronic properties of different molecular systems; however, in this work we have used the second one (renormalization process) due to its low computational cost compared to the first one (DFT), facilitating even more numerical or analytical calculations [31–36].

Returning to the purpose of this work, it should be noted that we focus on determining the thermoelectric properties of the planar diphenyl-ether molecule, taking into account different structural forms, which can provide interesting behaviors such as obtaining a high figure of merit ($ZT$), which depends on the electrical conductance ($\mathcal{G}$), the Seebeck coefficient ($S$), and the thermal conductance ($\kappa$). Now, to obtain a more extensive knowledge of

the $ZT$ behavior, these thermoelectric quantities are explored using the already known and standard method of tight binding, calculating the transmission probability ($T$) by Green's functions through the Fisher–Lee relation [37–39], while the thermoelectric quantities are evaluated by Landauer's theory [38,39].

Our paper is organized as follows. In Section 2, we present the model of the diphenyl-ether molecule based on a TB Hamiltonian. In Section 3, we describe the method used for the calculations of different thermoelectric quantities. Finally, we report our essential discoveries as conclusions in Section 5. In addition to this, an Appendix A is presented as a complement to explain in detail the renormalization method or process of the system under study.

## 2. Model

To study the transport properties of the diphenyl ether molecule ($C_{12}H_{10}O$), it is connected between two metal electrodes (Left-$L$ and Right-$R$) through the atomic site $i$, resulting in different structural configurations depending on the connections between $(i - L/R)$, as shown in Figure 1. In the tight-binding approximation, the total system can be represented by the Hamiltonian given by:

$$H = H_M + H_L + H_I,$$

(1)

where $H_M$ corresponds to the Hamiltonian of the molecule, which has the following form:

$$H_M = \sum_i t_i \left( c_i^\dagger c_{(i+1)} + c_{(i+1)}^\dagger c_i \right) + \sum_i E_i c_i^\dagger c_i,$$

(2)

where $c_i^\dagger$ and $c_i$ are the operators for creating and destroying an electron at site $i$. $E_i$ is the site energy for the carbon ($E_c$) or oxygen ($E_o$) atoms, $t_i$ is the hopping between the atoms, which can be $t_v$ for C−C coupling into the benzene molecule, and $t_w$, when the coupling is between the benzene molecules and the oxygen atom. The terms $H_L$ and $H_I$ of Equation (1) represent the Hamiltonian of the leads and the molecule-leads interaction, respectively, and are given by:

$$H_L = \sum_{k_L} \varepsilon_{k_L} d_{k_L}^\dagger d_{k_L} + \sum_{k_R} \varepsilon_{k_R} d_{k_R}^\dagger d_{k_R},$$

(3)

$$H_I = \sum_{k_L} \Gamma_L d_{k_L}^\dagger c_1 + \sum_{k_R} \Gamma_R d_{k_R}^\dagger c_N + h.c.,$$

(4)

here, the operator $d_{k_{L(R)}}^\dagger$ is the creation operator of an electron in a state $k_{L(R)}$ with energy $\varepsilon_{k_L}$; $\Gamma_{L(R)}$ is the coupling between each lead with the molecule, and *h.c.* is the complex conjugate of the Hamiltonian.

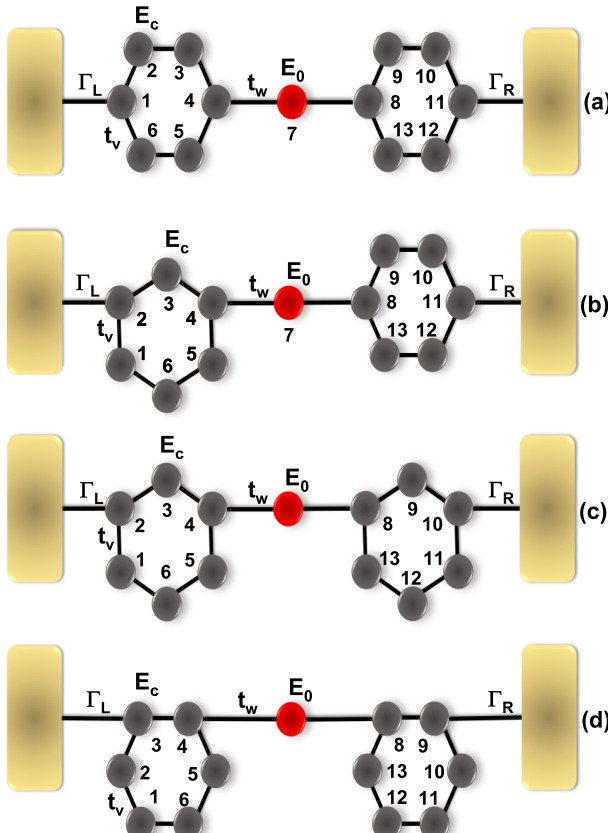

**Figure 1.** Diphenyl-ether molecule connected from different atomic sites to Left and Right leads. Model (a) Connections $1 - L$ and $11 - R$, Model (b) Connections $2 - L$ and $11 - R$, Model (c) Connections $2 - L$ and $10 - R$ and Model (d) Connections $3 - L$ and $9 - R$.

## 3. Method

To determine the thermoelectric properties of the diphenyl-ether molecule, we first focus on calculating the fundamental property that we need in order to determine the others, which is defined as the transmission probability $T(E)$ as a function of the energy with which the electron enters through the molecular system. This property is calculated using Green's function techniques, through the Fisher–Lee relation [37–39] and is given by:

$$T(E) = Tr[\Gamma_L G^r \Gamma_R G^a], \tag{5}$$

where $G^{a(r)}$ represents the advanced (retarded) Green's functions and $\Gamma_{L(R)}$ represents the spectral density matrices of the Left (Right) electrodes, which are given by $\Gamma_{L(R)} = i\left(\Sigma_{L(R)} - \Sigma_{L(R)}^\dagger\right)$ (were $\Sigma_L = \Sigma_R = -i\Gamma/2$). Now, to calculate the total Green functions given in the expression (5) for each model (see Appendix A for details on the calculation of Green's functions for each model using the renormalization scheme), the Dyson equation is used, taking into account that $G = G^0 + G^0(\Sigma_L + \Sigma_R)G$, where $G^0$ is the Green function of the molecular system without being coupled to the contacts. It is important to note that, as soon as the decimation process is performed for all models, and the systems are reduced to one-dimensional models, the Fisher–Lee relation for our calculations becomes:

$$T(E) = \Gamma_R \Gamma_L |G_{1N}|^2, \tag{6}$$

where $N$ is the number of sites in the new linear system.

Then, once the transmission probability is determined by relationship (6), we can analyze the antiresonances in the $T(E)$ profile for all models, where both the Green's function and transmission function are equal to zero, indicating that the cofactor is also

zero. However, not all zeros of the cofactor result in an antiresonance in the transmission; therefore, we calculate the cofactor where there is no antiresonance and are able to calculate the Green's function, as a function of the cofactor, using the expression:

$$G_{ij} = \frac{C_{ji}(E-H)}{det(E-H)},$$

(7)

where $C_{ij}$ is the cofactor of the hamiltonian matrix $H$ and is given by:

$$C_{ij}(H) = (-1)^{j+i} det(h_{ji}),$$

(8)

where $h_{ji}$ is a submatrix of $H$ obtained by removing the $j$th row and the $i$th column [40].

In particular, when there is a cyclic molecular system, such as a benzene ring, it is possible to convert the 6-site system to two effective sites, with a $2 \times 2$ effective Hamiltonian matrix having the following form:

$$H_{eff} = \begin{pmatrix} \tilde{E}_a & V_{ab} \\ V_{ab} & \tilde{E}_b \end{pmatrix},$$

(9)

where the diagonal terms ($E_a$ and $E_b$) are the effective energies of sites $a$ and $b$, respectively, which result from renormalizing the $n - sites$ with energy $E_i$ to an effective site, and off-diagonal terms are the effective couplings between the two effective sites mentioned; with this Hamiltonian, we calculate the Green's function from site $a$ to $b$ with Equation (7), given the expression:

$$G_{ab} = \frac{V_{ab}}{(E - \tilde{E}_a)(E - \tilde{E}_b) - V_{ab}^2}.$$

(10)

From Equation (10), we can observe that when $V_{ab}$ is zero, the Green's function is zero, and therefore, the profile in the transmission probability results in an anti-resonance [41].

Likewise, having the transmission calculated, the current flowing through the molecular system (which is considered as a process of dispersion of an electron between the contacts), is calculated using the Landauer formalism by means of the expression:

$$I(V) = I_0 \int_{-\infty}^{\infty} (f_L - f_R) T(E) dE,$$

(11)

where $I_0 = 2e/h$, $e$ is the electronic charge, $h$ represents the Plank's constant, and $f_{L(R)} = [1 + \exp(\beta(E - \mu_{L(R)}))]^{-1}$ is the Fermi–Dirac distribution function, where $\beta = 1/k_B \Theta$ is the Botzmann's factor, and $\mu_{L(R)} = E_f \pm eV/2$ is the chemical potential [41].

Additionally, we use the Landauer integrals $\mathcal{L}_n$, which have the following expression:

$$\mathcal{L}_n = -\int T(E)(E - E_f)^n \left( \frac{\partial f(E)}{\partial E} \right) dE,$$

(12)

where $E_f$ describes the equilibrium Fermi energy of the system under the zero bias condition. With this expression (Equation (7)), we calculate the thermoelectric properties (object of this work) such as the electrical conductance $\mathcal{G}$, the Seebeck coefficient $S$, the thermal conductance $\kappa$, and the figure of merit $ZT$ defined as:

$$\mathcal{G} = \frac{2e^2}{h} \mathcal{L}_0,$$

(13)

$$S = -\frac{1}{e\Theta} \frac{\mathcal{L}_1}{\mathcal{L}_0},$$

(14)

$$\kappa = \frac{2}{h\Theta} \left( \mathcal{L}_2 - \frac{\mathcal{L}_1^2}{\mathcal{L}_0} \right),$$

(15)

$$ZT = \frac{\mathcal{G}S^2\Theta}{\kappa} = \frac{\mathcal{L}_1^2}{\mathcal{L}_0\mathcal{L}_2 - \mathcal{L}_1^2}, \tag{16}$$

here, $\Theta$ is the equilibrium temperature [25].

## 4. Results and Discussion

In this section, we analyze the quantum transport properties ($T(E)$, $I(V)$, $\mathcal{G}$, $S$, $\kappa$, and $ZT$) through a single diphenyl-ether molecule, considering four different structural models (see Figure 1).

In the first instance, and in order to validate the real space renormalization method used in this work, a comparison of the calculation of the transmission probability determined for two molecular systems similar to that of diphenyl-ether is performed (see Figure 2). The first system taken into account is a 4,4′-diaminodiethyl diphenyl-ether molecule ($C_{12}H_{12}N_2O$), through which Wang Y.H et al. calculated the transmission using the DFT method. Such a molecule contains two benzene rings linked by an oxygen atom, two $NH_2$ groups that couple the molecule to the electrodes, and a second system that is also characterized by two benzene rings with two H atoms, which serve as a bridge to couple the molecule to the electrodes and is called bis-(4-mercaptophenyl)-ether ($C_{12}H_{10}S_2O$). For this last molecular system, the transmission is calculated analytically with the method used in this work, where the site energy used for the Sulfur atom $E_s = -0.98$ eV, the site energy of the Carbon atom $E_c = 0$ eV, the site energy for the Oxygen atom $E_o = -2.2$ eV. For the hopping values, we have S–C, $t_s = -0.83$ eV; C–C, $t_c = -2$ eV; and O–C, is $t_o = -1$ eV.

Figure 2 shows the probability of transmission as a function of the energy of the incident electron, for the 4,4′-diaminodiphenyl-ether systems calculated by DFT (blue curve), and the bis-(4-mercaptophenyl)-ether molecule by the analytical method (black curve), which is compared with the data obtained for the diphenyl-ether model (a) system (red curve). We observed that around the Fermi level ($E_F = 0$), the same behavior occurs, which is a destructive quantum interference in the transmission caused by the oxygen site, which breaks the delocalization of the system; this agrees with results from previous works [15]. On the other hand, Figure 2 shows that for the three systems, the energy of the HOMO and the LUMO are very similar, which are around $-1.25$ and $2.00$ eV, presenting a gap of approximately the same width. In order to find a good comparison with the system reported by DFT, a $\Gamma = 4$ eV was used for the bis-(4-mercaptophenyl)-ether system, and a $\Gamma = 0.5$ eV for the dyphenyl-ether system, which tells us that the additional radicals cause a stronger coupling with the electrodes, compared to the dyphenyl-ether molecule alone. It is important to note that the $\Gamma$-parameter for the three molecules is different, $\Gamma = 4$ eV corresponds to a strong coupling to the electrodes where the electrode molecular orbitals hybridize strongly with the molecule orbitals, while $\Gamma = 0.5$ eV corresponds to a weaker hybridization. In the DFT calculation, no report of $\Gamma$ was expelled. Furthermore, the magnitude difference around the Fermi level for the three curves was irrelevant since it accounts for very small transmission values. In the models shown in Figure 1, the site energies $Ec = 0$ eV and $E_o = -2.2$ eV were taken, as the potential between the C–C sites is $t_v = 1$ eV and C–O is $t_w = -1$ eV.

Figure 3 presents the transmission profile as a function of the incident electron energy, for a value of $\Gamma = 0.2$ eV (weak coupling).

The resonant peaks observed in the transmission plot are associated with the eigenvalues of the diphenyl ether molecule. On the other hand, 9 resonances are seen (degenerate and non-degenerate) in model (a), which are related to the 13 eigenvalues of the molecule ($-3.06, -2.00, -1.78, -1.00, -0.63, 1.00, 1.19, 2.00, 2.09$ eV); eigenvalues ($1.00, -1.00$ eV) are triple degenerate. When considering weak coupling, $\Gamma = 0.2$ eV, it implies that the electronic states of the electrodes have not mixed with those of the molecule. For this reason, there are quite defined peaks. Meanwhile, for all models, there are antiresonances with some eigenvalues of the molecule, associated with destructive quantum interference. This fact can be explained by means of Feynman integrals, where the electron can traverse

all possible paths within the molecule and, depending on the paths chosen, can arrive at the drain electrode in phase (constructive interference) or out of phase (destructive interference). For model (a), the antiresonance is at $E_F = 0$ eV; for model (b), at $E_F = 0$, $\pm 1$ eV; for model (c), at $E_F = 0$, $\pm 1$ eV; and for model (d), at $E_F = 0$, $\pm 1$ eV, $\pm 1.4$ eV. Furthermore, if these paths cancel in pairs, it leads to a node at the transmission probability at $E = E_F$ [42], as in the case of models (b) and (c).

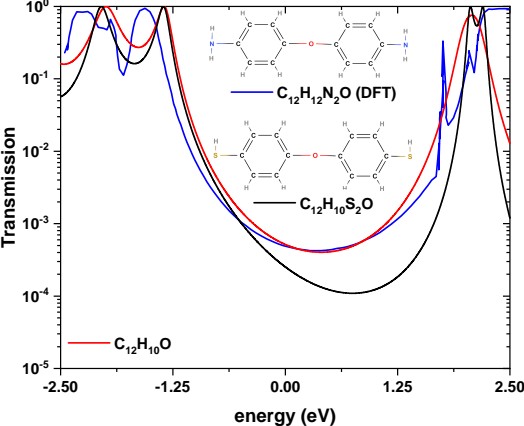

**Figure 2.** Transmission probability as a function of incident electron's energy for the molecular systems: 4,4′-diaminodiphenyl-ether (blue curve), bis-(4-mercaptophenyl)-ether (black curve), and dyphenyl-ether model (a) (red curve). For the bis-(4-mercaptophenyl)-ether molecule a $\Gamma = 4$ eV was used, and for the dyphenyl-ether molecule, a $\Gamma = 0.5$ eV was used.

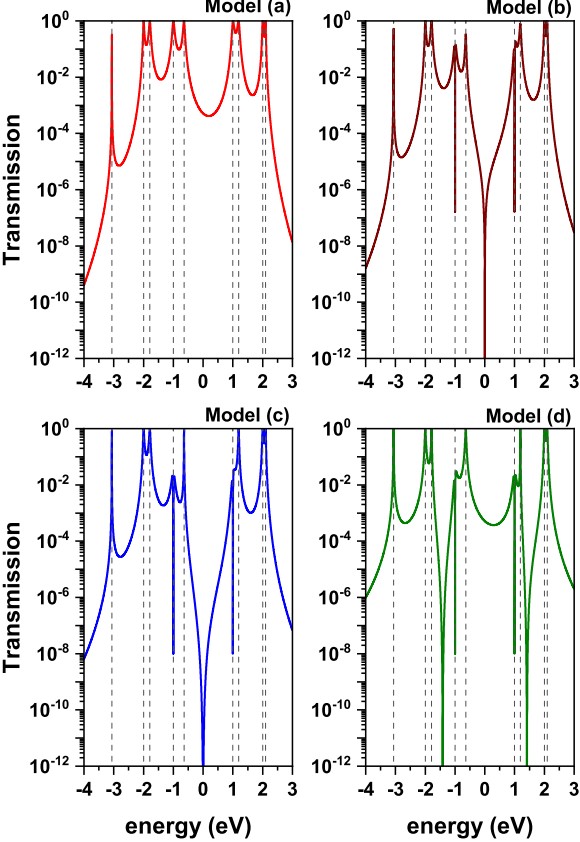

**Figure 3.** Transmission probability for the molecule diphenyl-ether, as a function of the energy of the incident electron, for models (a–d), which have different coupling sites with the electrodes. The calculations are in a weak coupling regime ($\Gamma = 0.2$ eV).

### 4.1. Cofactor

Figure 4 shows the graphs of the cofactor comparison with the Green functions, as well as the effective coupling to the two sites of all models (a, b, c, and d), in the weak coupling regime ($\Gamma$ = 0.2 eV). In model figure (a), there are two roots for the cofactor ($C_{11-1}$), but there are no crossings with the Green's function ($G_{11-1}$), nor with the effective coupling ($V_{11-1}$) on the cofactor zeros. This agrees with what is shown in the transmission plot of Figure 3 for model (a), in which we do not have destructive quantum interference. For models (b) and (c), there are three roots for $C_{11-2}$, and there are three crossings at these energy values, one for zero energy with $G_{11-2}$ (model (b)), and $G_{10-2}$ (model (c)), and two with energy $-1$ eV and 1 eV, with the $V_{11-2}$ and $V_{10-2}$. Comparing these results with the graph of the transmission of Figure 3 for models (b) and (c), it is expected that it presents the three destructive quantum interferences at 0, 1, and $-1$ eV. For model (d), we have a similar analysis, but this model presents four cofactor crossings, two with $G_{9-3}$, at energies of $-1.4$ and 1.4 eV, and two with $V_{9-3}$ at $-1$ and 1 eV, and this agrees with what is shown in Figure 3, where, for the d model, there are four destructive quantum interferences. This analysis allows us to verify the results obtained for the probability of transmission. The Green's function $G_{11-1}$ and the effective coupling $V_{11-1}$ represent model (a), where the molecule is connected to sites 1 and 11. On the other hand, $G_{11-2}$ and $V_{11-2}$ correspond to model (b), in which the molecule is connected to the electrodes of sites 2 and 11. Regarding model (c), $G_{10-2}$ and $V_{10-2}$ are used when the molecule is connected to the electrodes of sites 2 and 10. Finally, for the system connected through sites 9 and 3 (model (d)), $G_{9-3}$ and $V_{9-3}$ are employed.

### 4.2. Transmission by Varying $\Gamma$

Figure 5 shows a sweep in both $\Gamma$ and the energy of the incident electron for the diphenyl-ether molecular system, for different coupling sites of the molecule and the metal electrodes.

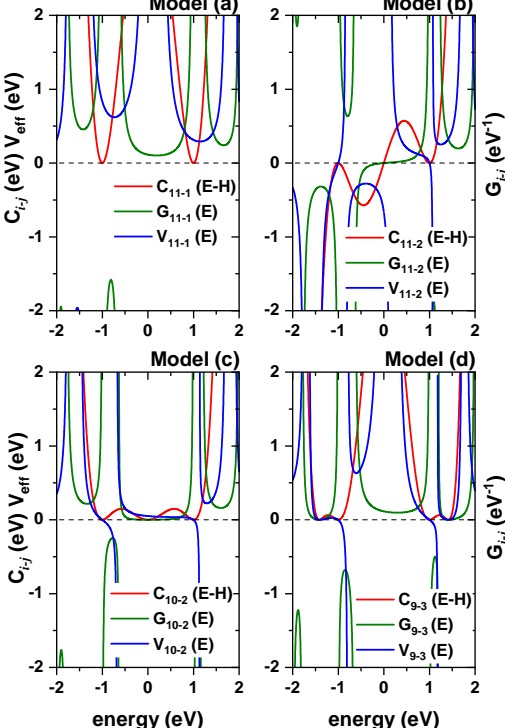

**Figure 4.** Comparison of the cofactor, with the Green's function of the different contact sites, and the effective potential at two sites of the models (a–d). In the weak coupling regime with the electrodes ($\Gamma$ = 0.2) eV, where $G_{i-j}$ and $C_{i-j}$, is the Green's function, and the cofactor of the system coupled to site $i$ and site $j$, respectively.

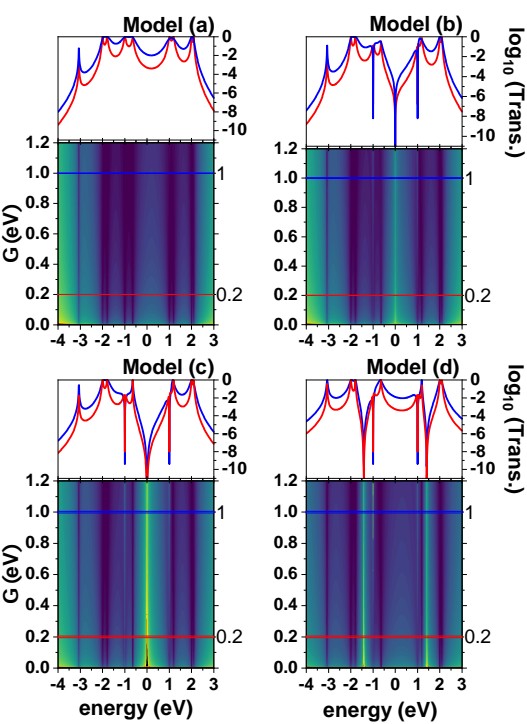

**Figure 5.** Transmission probability of the diphenyl-ether molecule, as a function of the energy of the incident electron and the coupling potential with the lead (Γ), for models (a–d). In the upper part, the red line corresponds to the logarithm of the transmission for a value of Γ = 0.2 eV (weak coupling), and the blue curve corresponds to the logarithm of the transmission for a value of Γ = 1 eV (strong coupling).

At the bottom of the plots is a contour plot of the transmission probability logarithm, on the vertical axis is the sweep in Γ, and on the horizontal axis is the sweep in the energy of the incident electron. In the upper part is the probability of the transmission logarithm as a function of the incident electron energy, for two different values of Γ, one of weak coupling (0.2 eV), and another for strong coupling (1 eV). For the strong coupling, in all the models the loss of some peaks is observed in the transmission probability graph (upper graph). This is due to the peaks overlapping since the Γ parameter is related to the peak at half maximum. This is due to the hybridization that exists between the delocalized states of the electrode and the localized states of the molecular system. For example, in model (a), for weak coupling (red graph) for energies of −2 eV, we have two close peaks; something similar happens for energies of −1, 1, and 2 eV, but when there is strong coupling, these two peaks combine, forming one. This same analysis can be completed for models (b)–(d).

### 4.3. Current-Voltage Characteristics

Figure 5 shows the plot of current versus voltage with the Fermi equilibrium energy $E_F$ set at 0 eV and a temperature of 300 K.

Figure 6 shows the plot of current versus voltage, the Fermi equilibrium energy $E_F$ was set at 0 eV and the temperature at 300 K. It is noticed that in regions of constant current, where the system is far from the transmission resonances (see Figure 3), and steep current regions where the transmission resonances are located. Furthermore, the current curves are antisymmetric with respect to zero volts. From Figure 6, it can be seen that as the voltage increases, the gap between the electrodynamic potentials of the electrodes (left and right) becomes larger, and when the value of the electrodynamic potential coincides with a value characteristic of the molecule, there is a jump in the injection energy of the electron, until reaching a saturation value, which is when the molecule cannot store more electrons. Hence, the injection energy becomes constant and there will be no more energy jumps, no matter how much the voltage is increased. This saturation value is reached for a voltage of

approximately 4 volts; at this point, a maximum current amplitude is obtained. Model (a) is the one with the greatest amplitude in the current, and is to be expected, since it presents the greatest area under the curve in the transmission graph, followed by model (b), model (c), and finally model (d). In the systems shown in the *I* vs. *V* graph, all present a value of 0 current in the voltage range of −1 volt and 1 volt, which indicates that the systems are not conductive, but it can be considered a semiconductor since they present this gap, which is due to the small transmission value seen in Figure 3. This feature can be useful for designing an electronic switch.

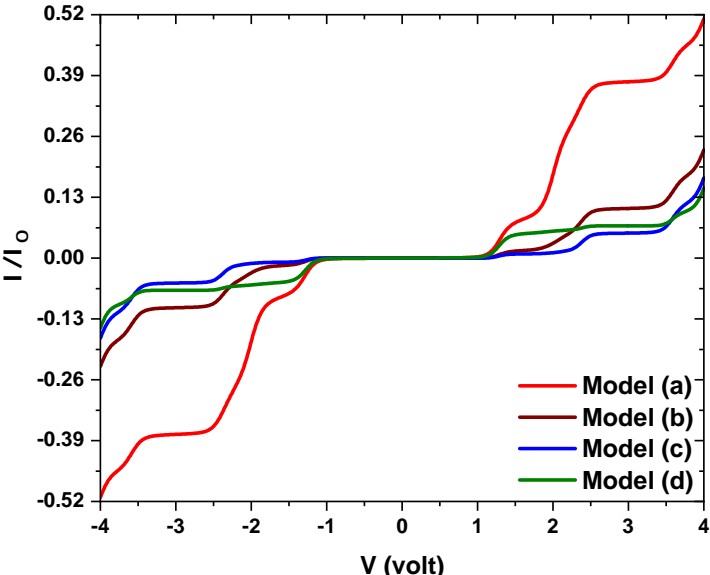

**Figure 6.** Current –voltage curve for a single diphenyl-ether molecule connected to two electrodes. model (a) (red line), model (b) (wine line), model (c) (blue line), and model (d) (green line).

*4.4. Electrical Conductance*

Figure 7 shows the electrical conductance of a single diphenyl ether molecule, as a function of the Fermi energy, for models (a)–(d), for 3 different temperatures 50 K (red), 100 K (blue), and 300 K (black). The calculations are in the weak coupling regime ($\Gamma = 0.2$ eV); that is, the behavior of the molecule will be studied when the electrons pass through it. For all the models shown in the electrical conductance graph, it can be seen that it is proportional to the transmission probability (see Figure 3 and Equation (13)); that is, they share the same number of resonant peaks in the same energy positions. In all models, it can be noted that a forbidden band is generated, which is between −0.63 eV and 1 eV, due to destructive quantum interference between the localized states of the molecule and the delocalized states of the electrodes. At first glance, not many changes in conductance are noticeable for the different temperatures, but in the inset image of the model (a), it can be seen that the conductance decreases as the temperature increases. This decrease is a consequence of the derivative decrease, with respect to the energy, of the Fermi-Dirac distribution function in the integrand of the conductance Equation (13), as the temperature increases (see the insert of Figure 7 corresponding to model (b)).

The decrease in conductance is also related to scattering by molecular vibrations, the mechanism is not included in the present work. This occurs due to the fact that when the temperature increases, there is an increase in the electron's kinetic energy since they move faster, thus increasing the amplitude of vibration of the atoms of the molecular system, thus behaving as a harmonic oscillator. This phenomenon causes the resistance to the passage of free electrons to increase since it is difficult to pass from one electrode to the other since the interference of the atoms with the trajectories of the valence electrons throughout the molecule increases.

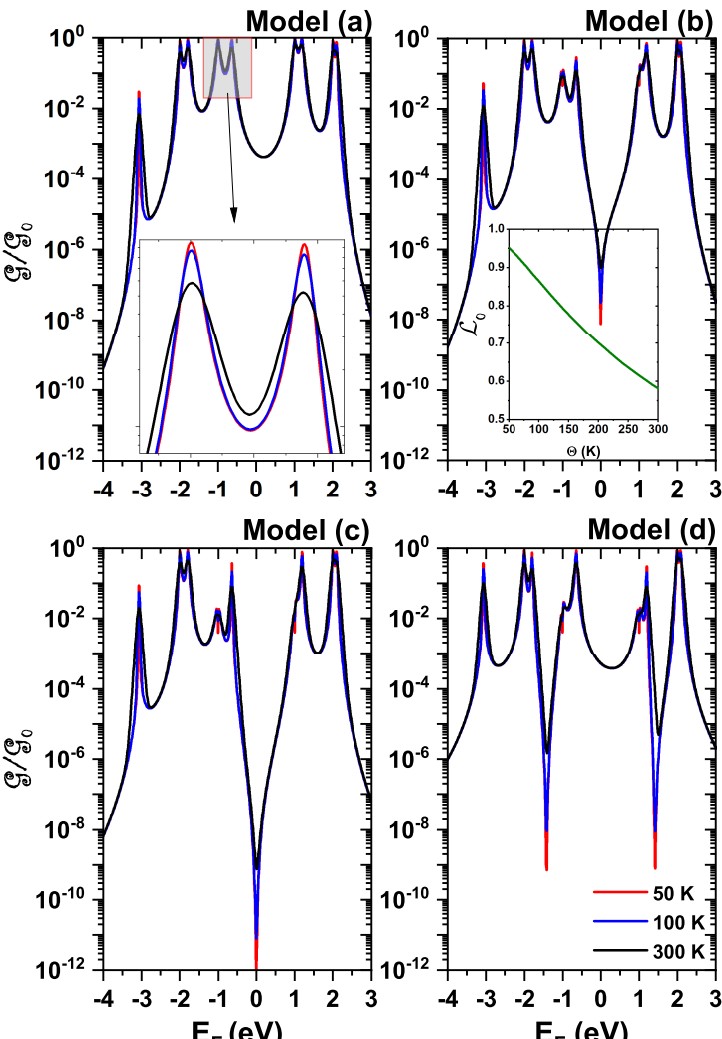

**Figure 7.** Electrical conductance in a single diphenyl-ether molecule, as a function of the Fermi energy, for models (a–d), which have different coupling sites with the electrodes. The graph shows calculations for 3 different temperatures 50 K (red), 100 K (blue), and 300 K (black). The calculations are in a weak coupling regime ($\Gamma = 0.2$ eV). The inset in the figure corresponding to model (a) is an enlargement of the shadow region at the upper part of the figure. The insert in the figure corresponding to model (b) is the graph of the integral $\mathcal{L}_0$ as a function of temperature.

### 4.5. Seebeck Coefficient

The thermoelectric properties are now shown, starting with the Seebeck coefficient, then the thermal conductance, and finally, the $ZT$ or figure of merit, taking into account the calculations of the previous physical quantities, to determine the efficiency of the models to convert thermal energy into electricity.

Figure 8 shows the Seebeck coefficient ($S$) in a single molecule of diphenyl-ether, as a function of the Fermi energy, for models (a)–(d). The graph shows calculations for 3 different temperatures 50 K (red), 100 K (blue), and 300 K (black). The calculations are in a weak coupling regime ($\Gamma = 0.2$ eV). The $S$ is extracted from the results close to the HOMO−LUMO gap. Table 1 shows parameters taken from Figures 3, 7 and 8, for 300 K. It is observed that the Fermi energy is closer to the HOMO level, indicating a positive $S$ value for all models considered. Furthermore, it is verified that high $S$ values are coupled with low conductance values. The most notable result is that the $S$ is higher for models (b) and (c) where an antiresonance occurs at the Fermi energy, which coincides with the low HOMO conductance. It is important to note that the low temperature Seebeck coefficient results

are very small due to the $\mathcal{L}_0$ (Equation (12)), increasing magnitude as the temperature decreases (see the insert of Figure 7 corresponding to model (b)).

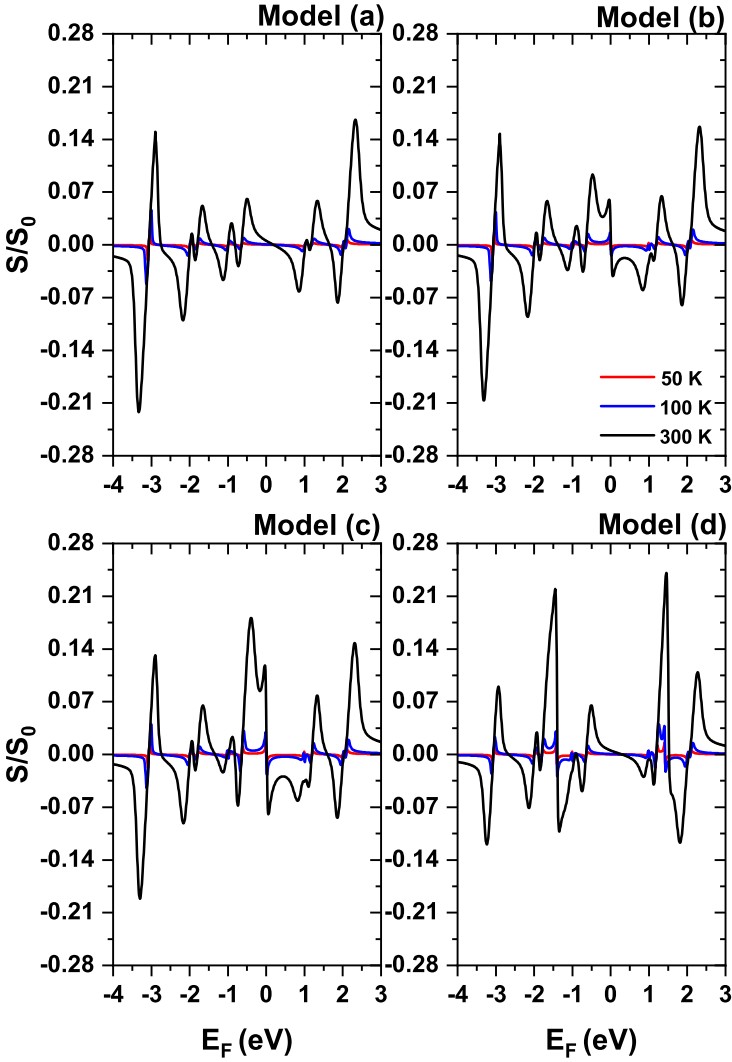

**Figure 8.** Seebeck coefficient on a single diphenyl-ether molecule, as a function of Fermi energy, for models (a–d), which have different docking sites with the electrodes. The graph shows calculations for 3 different temperatures 50 K (red), 100 K (blue), and 300 K (black). The calculations are in a weak coupling regime ($\Gamma = 0.2$ eV).

**Table 1.** Seebeck coefficients. h = high, l = low, H = HOMO, L = LUMO.

| Model | H | L | Conductance | $S/S_0$ | *ZT* |
|---|---|---|---|---|---|
| **a** | −0.63 | 1.00 | h-H, h-L | 0.06 | 1.59 |
| **b** | −0.63 | 1.00 | l-H, h-L | 0.09 | 3.85 |
| **c** | −0.63 | 1.00 | l-H, h-L | 0.18 | 25.78 |
| **d** | −0.63 | 1.00 | h-H, l-L | 0.06 | 2.09 |

### 4.6. Thermal Conductance

Now, we show the thermal conductance as a function of the Fermi energy, for a weak coupling of 0.2 eV, and for different temperatures.

Figure 9 shows that thermal conductance increases as temperature increases, unlike electrical conductance. This is due to the fact that the increase in temperature, the integral

of $\mathcal{L}_2$ and $\mathcal{L}_1$, increases in value (see inset in the Figure 9 corresponding to the model (d)). From Equation (15), the electrical conductance depends on the sum of $\mathcal{L}_2$ and the division of $\mathcal{L}_1^2$ by $\mathcal{L}_0$, although the latter decreases with temperature, gain $\mathcal{L}_1^2$. Another way of looking at thermal conductance is directly related to the increase in vibration of the atoms in the molecule, and these vibrations increase the average speed of the particles and therefore increase heat transfer, which is related to thermal conductance of the proposed models. The one with the greatest amplitude in thermal conductance is (a); this is related to the symmetry this model presents since the electrons leave one electrode to another. This is because when dividing the molecule, like the molecule travels, they recombine through the same number of sites, arriving in phase (see Figure 3).

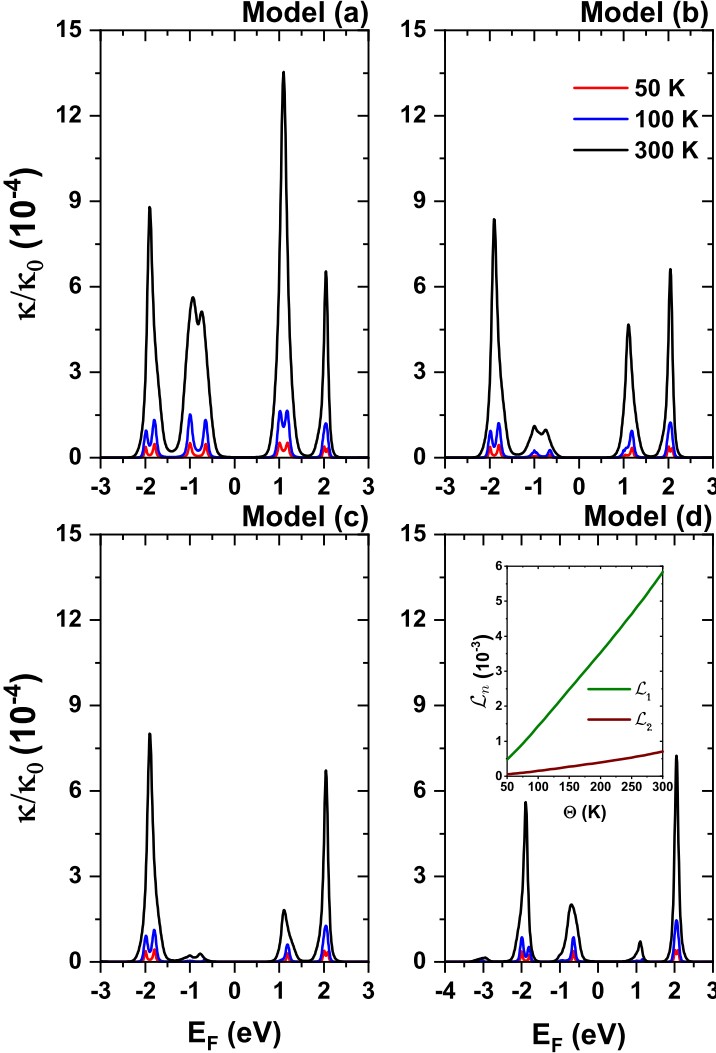

**Figure 9.** Thermal conductance in a single diphenyl-ether molecule as a function of Fermi energy for models (a–d), which have different coupling sites with the electrodes. The graph shows calculations for 3 different temperatures 50 K (red), 100 K (blue), and 300 K (black). The calculations are in a weak coupling regime ($\Gamma = 0.2$ eV). The insert in the figure corresponding to model (d) is the graph of the integral $\mathcal{L}_1$, and $\mathcal{L}_2$ as a function of temperature.

### 4.7. Figure of Merit ZT

Finally, we have in Figure 10 the $ZT$, or figure of merit, shown as a function of the Fermi energy ($E_F$) to carry out the calculations of the $ZT$ the results obtained for $G$, $S$, and $\kappa$. Of the figures for the $ZT$, the model that presents the highest value is (c), which is consistent with the calculations obtained for the Seebeck Coefficient (see Table 1), since it is

the model that presents the highest *S*, and the higher the *S*, the higher the value of *ZT*. This is due to the fact that model (c) presents an interference in the graph of the transmission probability around the Fermi level, and this is more pronounced than in model (b), which is the model with the second highest *ZT*.

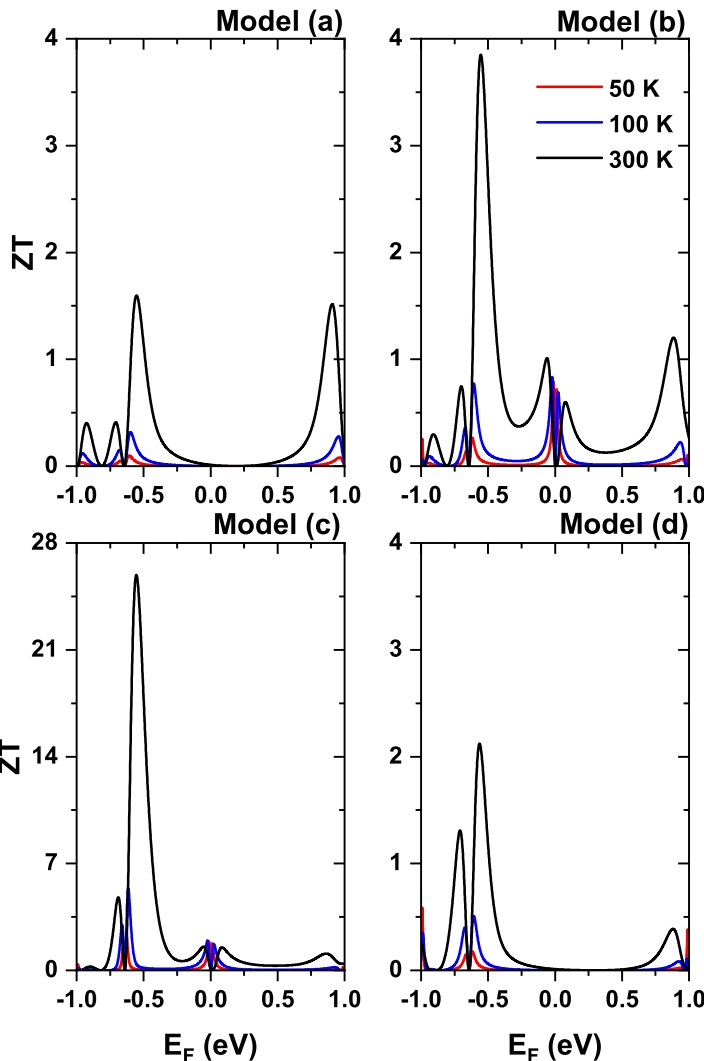

**Figure 10.** *ZT* figure of merit in a single diphenyl-ether molecule, as a function of the Fermi energy, for models (a–d), which have different binding sites. Coupled with the electrodes, the graph shows calculations for 3 different temperatures 50 K (red), 100 K (blue), and 300 K (black). The calculations are in a weak coupling regime $\Gamma = 0.2$ eV.

The model with the smallest *ZT* is presented by (a); therefore, it is with the smallest *S*, which is the most symmetrical model, and does not present interference in its transmission. For all the models shown, the $ZT \geq 1$. These values indicate that the systems studied can present a good efficiency in the conversion of thermoelectric energy, which is slightly higher than the *ZT* value of the commercially available inorganic semiconductor, bismuth telluride ($Bi_2Te_3$), which has a *ZT* of 1 [43]. Lastly, it is important to mention that the higher the temperature, the higher the efficiency.

## 5. Conclusions

The electrical and thermoelectric properties of a single molecule of diphenyl ether, connected to two metal electrodes, were studied for different atomic sites of the molecule with the electrodes. Using the technique of Green's functions out of equilibrium to renormalize

the molecule to five effective sites, and with the Landauer–Büttiker formalism, the electrical and thermoelectric properties were calculated, finding very appreciable changes in the thermoelectric properties when taking into account different coupling sites between the molecular system and leads. For example, that the transmission graph for Model (a) does not present quantum antiresonances, as the other models do, this behavior makes this model present greater amplitude in the current graph. The model that presents the best behavior in thermoelectric properties is (c), since it presents the highest value in the Seebeck coefficient, and therefore, presents the highest $ZT$, or figure of merit. Lastly, it is worth mentioning that all of the models studied would be great candidates for thermoelectric devices since the value of $ZT$ is greater than or equal to 1.

**Author Contributions:** R.G.T.-N.: Conceptualization, methodology, software, formal analysis, investigation, writing; J.C.L.-G. and J.A.V.: Methodology, software; A.L.M.: Formal analysis, investigation, supervision, writing; J.H.O.S. and C.A.D.: Formal analysis, writing. All authors have read and agreed to the published version of the manuscript.

**Funding:** The authors are grateful to the Colombian Agencies: CODI-Universidad de Antioquia (Estrategia de Sostenibilidad de la Universidad de Antioquia and projects "Propiedades magneto-ópticas y óptica no lineal en superredes de Grafeno", "Estudio de propiedades ópticas en sistemas semiconductores de dimensiones nanoscópicas", "Propiedades de transporte, espintrónicas y térmicas en el sistema molecular ZincPorfirina", and "Complejos excitónicos y propiedades de transporte en sistemas nanométricos de semiconductores con simetría axial"), and Facultad de Ciencias Exactas y Naturales-Universidad de Antioquia (ALM and CAD exclusive dedication projects 2022–2023).

**Data Availability Statement:** No new data were created nor analyzed in this study. Data sharing is not applicable to this article.

**Acknowledgments:** J.H.O.S. Acknowledges to the Centro de Gestión de Investigación y Extensión de la Facultad de Ciencias CIEC-UPTC-Tunja.

**Conflicts of Interest:** The authors declare no conflict of interest.

## Appendix A

### Appendix A.1. Decimation Procedure

We are going to study the quantum transport through the Diphenyl-ether molecule, changing the coupling site of the molecule with the electrodes, in total 4 models will be studied, see Figure 1, we will also show generalities of the decimation process of all the models.

### Appendix A.2. Decimation Model (a)

To perform the decimation of model (a), first the renormalization for the benzene rings to two effective sites is performed:

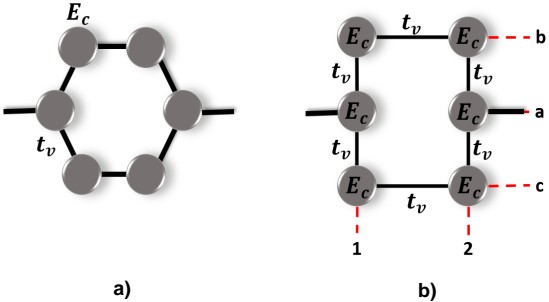

**Figure A1.** First benzene ring of the model (a), (**a**) isolated molecule, (**b**) reorganization of the molecule to carry out the renormalization.

Figure A1a shows the isolated benzene ring, and Figure A1b shows a reorganization of the ring in order to perform the renormalization, where the information enters the ring

through site $1a$ and exits through site $2a$, applying the Dyson equation to first neighbors, the following equations are obtained.

$$G_{11}^a = g_c + g_c t_v G_{11}^b + g_c t_v G_{11}^c, \tag{A1}$$

$$G_{11}^b = g_c t_v G_{11}^a + g_c t_v G_{12}^b, \tag{A2}$$

$$G_{11}^c = g_c t_v G_{11}^a + g_c t_v G_{12}^c, \tag{A3}$$

$$G_{12}^c = g_c t_v G_{11}^c + g_c t_v G_{12}^a, \tag{A4}$$

$$G_{12}^b = g_c t_v G_{11}^b + g_c t_v G_{12}^a, \tag{A5}$$

Solving the system of equations gives an expression for $G_{11}^a$ in terms of $G_{12}^a$.

$$G_{11}^a = \frac{g_c(1 - g_c^2 t_v^2)}{1 - 3g_c^2 t_v^2} + \frac{2g_c^2 t_v^3}{1 - g_c^2 t_v^2} G_{12}^a, \tag{A6}$$

The above expression has the form of a Dyson equation, where the first term is the effective Green's function ($g_1$) of a site 1 connected to a site 2, the second term, which multiplies $G_{12}^a$, is the effective coupling ($v_1$) with the effective site 2 ($G_{11}^a = g_1 + g_1 v_1 G_{12}^a$).

$$g_1 = \frac{g_c(1 - g_c^2 t_v^2)}{1 - 3g_c^2 t_v^2}, \tag{A7}$$

and

$$v_1 = \frac{2g_c^2 t_v^3}{1 - g_c^2 t_v^2}, \tag{A8}$$

The above process works for both benzene rings, therefore, model (a) is decimated to a system with 5 sites, of which 4 are effective (see Figure A4a).

*Appendix A.3. Decimation Model (b)*

Figure A2a shows the isolated benzene ring, and Figure A2b shows a reorganization of the ring in order to perform the renormalization, where the information enters the ring through site $1a$ and exits through site $2b$, applying the Dyson equation to first neighbors, the following equations are obtained.

$$G_{11}^a = g_c + g_c t_v G_{11}^b + g_c t_v G_{11}^c, \tag{A9}$$

$$G_{11}^b = g_c t_v G_{11}^a + g_c t_v G_{12}^b, \tag{A10}$$

$$G_{11}^c = g_c t_v G_{11}^a + g_c t_v G_{12}^c, \tag{A11}$$

$$G_{12}^c = g_c t_v G_{11}^c + g_c t_v G_{12}^a, \tag{A12}$$

$$G_{12}^a = g_c t_v G_{12}^c + g_c t_v G_{12}^b, \tag{A13}$$

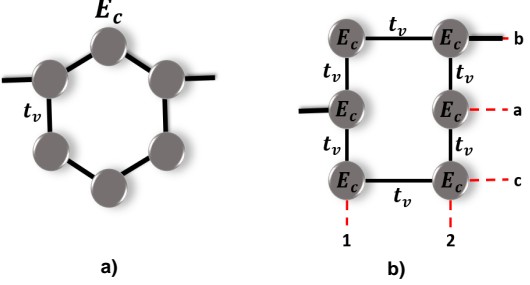

**Figure A2.** First benzene ring of the model (b), (**a**) isolated molecule, (**b**) reorganization of the molecule to carry out the renormalization.

Solving the system of equations gives an expression for $G_{11}^a$ in terms of $G_{12}^b$.

$$G_{11}^a = \frac{g_c(1 - 2g_c^2 t_v^2)}{1 - 4g_c^2 t_v^2 + 3g_c^4 t_v^4} + \frac{g_c t_v^2(1 - g_c^2 t_v^2)}{1 - 2g_c^2 t_v^2} G_{12}^b, \tag{A14}$$

The above expression has the form of a Dyson equation, where the first term is the effective Green's function ($g_2$) of a site 1 connected to a site 2, the second term, which multiplies $G_{12}^b$, is the effective coupling ($v_2$) with the effective site 2 ($G_{11}^a = g_2 + g_2\, v_2\, G_{12}^b$).

$$g_2 = \frac{g_c(1 - 2g_c^2 t_v^2)}{1 - 4g_c^2 t_v^2 + 3g_c^4 t_v^4}, \tag{A15}$$

and

$$v_2 = \frac{g_c t_v^2(1 - g_c^2 t_v^2)}{1 - 2g_c^2 t_v^2}, \tag{A16}$$

The process carried out for the first ring of the model (a) serves for the second ring of the model (b), therefore, the model (b) is decimated to a system with 5 sites, of which 4 are effective, two with effective Green's functions $g_1$ and effective coupling $v_1$, and two with effective Green's functions $g_2$ and effective coupling $v_2$ (see Figure A4b).

### Appendix A.4. Decimation Model (c)

For model (c), the two rings have the same configuration as the first ring of model (b), therefore, when performing the decimation process, the effective Green's function $g_2$ is obtained, and the effective coupling $v_2$, after the renormalization process, the model (c) is decimated to a linear system of 5 sites, where 4 are effective (see Figure A4a).

### Appendix A.5. Decimation Model (d)

Figure A3a shows the isolated benzene ring, and Figure A3b shows a reorganization of the ring in order to perform the renormalization, where the information enters the ring through site 1b and exits through site 2b, applying the Dyson equation to first neighbors, the following equations are obtained.

$$G_{11}^b = g_c + g_c t_v G_{11}^a + g_c t_v G_{12}^b, \tag{A17}$$

$$G_{11}^a = g_c t_v G_{11}^b + g_c t_v G_{11}^c, \tag{A18}$$

$$G_{11}^c = g_c t_v G_{11}^a + g_c t_v G_{12}^c, \tag{A19}$$

$$G_{12}^c = g_c t_v G_{11}^c + g_c t_v G_{12}^a, \tag{A20}$$

$$G_{12}^a = g_c t_v G_{12}^c + g_c t_v G_{12}^b, \tag{A21}$$

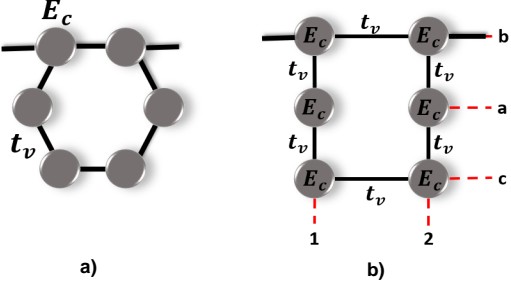

**Figure A3.** First benzene ring of the model (d), (**a**) isolated molecule, (**b**) reorganization of the molecule to carry out the renormalization.

Solving the system of equations gives an expression for $G_{11}^b$ in terms of $G_{12}^b$.

$$G_{11}^b = \frac{g_c(1 - 3g_c^2 t_v^2 + g_c^4 t_v^4)}{1 - 4g_c^2 t_v^2 + 3g_c^4 t_v^4} + \frac{t_v(1 - 3g_c^2 t_v^2 + 2g_c^4 t_v^4)}{1 - 3g_c^2 t_v^2 + g_c^4 t_v^4} G_{12}^b, \tag{A22}$$

The above expression has the form of a Dyson equation, where the first term is the effective Green's function ($g_3$) of a site 1 connected to a site 2, the second term, which multiplies $G_{12}^b$, is the effective coupling ($v_3$) with the effective site 2 ($G_{11}^b = g_3 + g_3 v_3 G_{12}^b$).

$$g_3 = \frac{g_c(1 - 3g_c^2 t_v^2 + g_c^4 t_v^4)}{1 - 4g_c^2 t_v^2 + 3g_c^4 t_v^4}, \tag{A23}$$

and

$$v_3 = \frac{t_v(1 - 3g_c^2 t_v^2 + 2g_c^4 t_v^4)}{1 - 3g_c^2 t_v^2 + g_c^4 t_v^4}. \tag{A24}$$

The above process works for both benzene rings, therefore, model (d) is decimated to a system with 5 sites, of which 4 are effective (see Figure A4a).

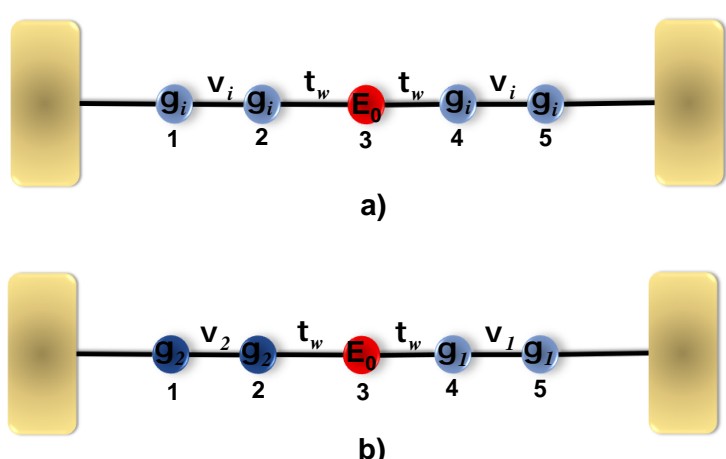

**Figure A4.** The models in Figure 1 were decimated to 5 sites, (**a**) models (a), (b) and (d), (**b**) model (d). *i* can take the values 1, 2 and 3, corresponding to models (a), (c) and (d), respectively.

All the models in Figure 1 are decimated to 5 sites, where we have 4 effective sites and the central oxygen site. For models (a), (c) and (d) they are symmetrical systems (see Figure A4a), where $g_i$ is the effective local Green's function and $v_i$ is the effective potential between sites, where *i* can be 1 (model (a)), 2 (model (c)) and 3 (model (d)). For model (b), we have an asymmetric system (see Figure A4b), which is a combination of the decimation made for the benzene rings of model a and the rings of model (c).

To obtain the Green's functions of sites 1 and 5 of the linear system, shown in Figure A4a, the Dyson equation is applied to first neighbors, we have:

$$G_{11}^{a,c,d} = g_i + g_i v_i G_{12}^{a,c,d}, \tag{A25}$$

$$G_{12}^{a,c,d} = g_i v_i G_{11}^{a,c,d} + g_i t_w G_{13}^{a,c,d}, \tag{A26}$$

$$G_{13}^{a,c,d} = g_o t_w G_{12}^{a,c,d} + g_o t_w G_{14}^{a,c,d}, \tag{A27}$$

$$G_{14}^{a,c,d} = g_i t_w G_{13}^{a,c,d} + g_i v_i G_{15}^{a,c,d}, \tag{A28}$$

$$G_{15}^{a,c,d} = g_i v_i G_{14}^{a,c,d}, \tag{A29}$$

Solving the system of equations gives the expression of the Green's functions $G_{11}^{a,c,d}$, and $G_{15}^{a,c,d}$. Due to the symmetry that the system in the Figure A4a presents, it can be affirmed that the functions of the sites $G_{11}^{a,c,d}$ and $G_{55}^{a,c,d}$ are equal and have the following form:

$$G_{11}^{a,c,d} = \frac{g_i(1 - g_i^2 v_i^2 - 2g_i g_o t_w^2 + g_i^3 g_o v_i^2 t_w^2)}{(1 - g_i^2 v_i^2)(1 - g_i^2 v_i^2 - 2g_i g_o t_w^2)}, \tag{A30}$$

we can also say the same about Green's functions $G_{15}^{a,c,d}$ and $G_{51}^{a,c,d}$, which are the same, which is:

$$G_{15}^{a,c,d} = \frac{g_i^4 g_o v_i^2 t_w^2}{(1 - g_i^2 v_i^2)(1 - g_i^2 v_i^2 - 2g_i g_o t_w^2)}. \tag{A31}$$

To obtain the Green's functions of the linear system shown in Figure A4b, the Dyson equation is applied to first neighbors, this system does not have a symmetrical shape, like the one shown in Figure A4a, for which the analysis is made in the direction 1 to 5, and from 5 to 1. Analyzing from site 1 to 5, we have:

$$G_{11}^b = g_2 + g_2 v_2 G_{12}^b, \tag{A32}$$

$$G_{12}^b = g_2 v_2 G_{11}^b + g_2 t_w G_{13}^b, \tag{A33}$$

$$G_{13}^b = g_o t_w G_{12}^b + g_o t_w G_{14}^b, \tag{A34}$$

$$G_{14}^b = g_1 t_w G_{13}^b + g_1 v_1 G_{15}^b, \tag{A35}$$

$$G_{15}^b = g_1 v_1 G_{14}^b. \tag{A36}$$

Analyzing from site 5 to 1, we have:

$$G_{55}^b = g_1 + g_1 v_1 G_{54}^b, \tag{A37}$$

$$G_{54}^b = g_1 v_1 G_{55}^b + g_1 t_w G_{53}^b, \tag{A38}$$

$$G_{53}^b = g_o t_w G_{54}^b + g_o t_w G_{52}^b, \tag{A39}$$

$$G_{52}^b = g_2 t_w G_{53}^b + g_2 v_2 G_{51}^b, \tag{A40}$$

$$G_{51}^b = g_2 v_2 G_{52}^b, \tag{A41}$$

Solving the system of equations gives the expressions of the Green's functions $G_{11}^b$, $G_{55}^b$, and $G_{15}^b$, and have the following form:

$$G_{11}^b = \frac{g_2(1 - g_1^2 v_1^2 - g_o g_1 t_w^2 - g_o g_2 t_w^2(1 - g_1^2 v_1^2))}{(1 - g_1^2 v_1^2)(1 - g_2^2 v_2^2) - g_o t_w^2(g_2 + g_1(1 - g_1 g_2 v_1^2 - g_2^2 v_2^2))}, \tag{A42}$$

$$G_{55}^b = \frac{g_1(1 - g_2^2 v_2^2 - g_o g_2 t_w^2 - g_o g_1 t_w^2(1 - g_2^2 v_2^2))}{(1 - g_1^2 v_1^2)(1 - g_2^2 v_2^2) - g_o t_w^2(g_2 + g_1(1 - g_1 g_2 v_1^2 - g_2^2 v_2^2))}, \tag{A43}$$

and

$$G_{15}^b = \frac{g_o g_1^2 g_2^2 v_1 v_2 t_w^2}{(1 - g_1^2 v_1^2)(1 - g_2^2 v_2^2) - g_o t_w^2(g_2 + g_1(1 - g_1 g_2 v_1^2 - g_2^2 v_2^2))}. \tag{A44}$$

With the effective Green's functions calculated $G_{11}^{a,b,c,d}$, $G_{15}^{a,b,c,d}$, and $G_{55}^b$, the total Green function of the system can be represented, given by the following expression:

$$G_{15} = \frac{G_{15}^{a,b,c,d}}{(1 - \Sigma_L G_{11}^{a,b,c,d})(1 - \Sigma_R G_{55}^{a,b,c,d}) - \Sigma_L \Sigma_R (G_{15}^{a,b,c,d})^2}. \tag{A45}$$

The new transmission probability for the effective molecular system in one dimension, as shown in Figure 2, can be written as:

$$T(E) = \Gamma_R \Gamma_L |G_{15}|^2. \tag{A46}$$

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
