# Peer review of "Theoretical Study of Thermoelectric Properties of a Single Molecule of Diphenyl-Ether"

_condensedmatter, doi:10.3390/condmat8030055_

Round 1

Reviewer 1 Report

The authors investigate the thermoelectric properties of a single molecule of Diphenyl-ether.

They successfully proposed models, provided methods for their computational approach. Moreover, they chose variety of adequate, and essential, computed measurements, being Electrical conductivity, Seebeck, and thermal conductivity and demonstrated and discussed their relation and correlation and finally produced ZT as thermoelctric efficiency.

The manuscript is very well structured, and English is generally understandable, and easy to follow. They adequately explained, showed, and reasoned the results and achieved interesting data especially ZT > 1. They could soundly explain the obtained simulation results, establishing an adequate reasoning for the physical phenomena governing them.

Nevertheless, this highly scientific interest to be published some minor concerns should be addressed and revision should be done:

1)   Although most of the paper English is well written in Abstract and introduction a small revision of the lines should be done taking into account the correct usage of points and comma as well as more scientific word choosing.

2)   As the manuscript is regarding the simulation and modeling of the thermoelectric properties of a single molecule of Diphenyl-ether the Title and keyword should be accordingly rectified in order to avoid confusion and to separate work from experimental ones.

3)    Regarding the author claim in various lines of the paper expressing ZT = 1 means 20 % higher than Carnot machine

In line 11 the authors written: “it exhibits an efficiency of approximately 20% compared to the Carnot machine”

While In line 313-316 and 329 “efficiency in the generation of thermoelectric energy is greater than 20% compared to the efficiency of the Carnot engine”.

So which one is correct? The author should explain this more while providing a reference especially where ZT is driven from Carnot machine, meaning maximum efficiency in ideal thermodynamic cycle.

Although most of the paper English is well written in Abstract and introduction a small revision of the lines should be done taking into account the correct usage of points and comma as well as more scientific word choosing.

Author Response

Reviewer 1:

Comments:
The authors investigate the thermoelectric properties of a single molecule of Diphenyl-ether.

They successfully proposed models, provided methods for their computational approach. Moreover, they chose variety of adequate, and essential, computed measurements, being Electrical conductivity, Seebeck, and thermal conductivity and demonstrated and discussed their relation and correlation and finally produced ZT as thermoelectric efficiency.

The manuscript is very well structured, and English is generally understandable, and easy to follow. They adequately explained, showed, and reasoned the results and achieved interesting data especially ZT > 1. They could soundly explain the obtained simulation results, establishing an adequate reasoning for the physical phenomena governing them.

Nevertheless, this highly scientific interest to be published some minor concerns should be addressed and revision should be done:

Issue 1.1: Although most of the paper English is well written in Abstract and introduction a small revision of the lines should be done considering the correct usage of points and comma as well as more scientific word choosing.

Response 1.1: The authors would like to express our most sincere gratitude to the Referee for his/her evaluation of our article. His/her opinions and comments have helped us to substantially improve our work.

Following the recommendations suggested by the referee, a review and correction has been made in the abstract, and introduction.

Issue 1.2: As the manuscript is regarding the simulation and modeling of the thermoelectric properties of a single molecule of Diphenyl-ether the Title and keyword should be accordingly rectified in order to avoid confusion and to separate work from experimental ones.

Response 1.2: Here the referee is right, therefore it has been decided to change the name to "Theoretical study of thermoelectric properties of a single molecule of Diphenyl-ether."

Issue 1.3: Regarding the author claim in various lines of the paper expressing ZT = 1 means 20 % higher than Carnot machine

In line 11 the authors written: “it exhibits an efficiency of approximately 20% compared to the Carnot machine”

While In line 313-316 and 329 “efficiency in the generation of thermoelectric energy is greater than 20% compared to the efficiency of the Carnot engine”.

So which one is correct? The author should explain this more while providing a reference especially where ZT is driven from Carnot machine, meaning maximum efficiency in ideal thermodynamic cycle.

Response 1.3: For this question, the referee's suggestion was taken into consideration, which implies not comparing our molecular system with a Carnot machine, the ideal maximum efficiency, but instead, the comparison is made with a very commercial inorganic thermoelectric material, bismuth telluride, which has a ZT equal to 1.

Reviewer 2 Report

The manuscript by R.G. Toscano-Negrette et al. reports computational electrical and thermal transport properties of the dyphenil ether molecule. The main finding is high ZT, however, the authors didn't describe, how they obtained the exact values of ZT from the Green functions-based calculations. Therefore, a major revision should be applied.

1) Figure 2: “For the Dyphenyl-ether molecule a G = 4 eV was used, and for the Dyphenyl-ether molecule a G = 0.5 eV was used”. Please, correct this misprint, and specify correct Gamma.

2) Figure 4: Correct the legend, “111” should be “11-1” etc.

3) Figure 5: The right upper Y axis should be logarithmic. Please, add a color legend explaining the bottom part of each panel.

4) Figure 6: Such potentials as +/-6V are high enough for the oxidation processes. The real system will not survive in these conditions.

5) Figure 10: add ZT values at E=EF for all models.

6) Calculation of ZT requires absolute values of electrical, thermal conductance, and Seebeck coefficient. However, Figures 7-9 present only relative values. Explain exactly, how the ZT values have been obtained.

Author Response

Reviewer 2:

Comments:
The manuscript by R.G. Toscano-Negrette et al. reports computational electrical and thermal transport properties of the dyphenil ether molecule. The main finding is high ZT, however, the authors didn't describe, how they obtained the exact values of ZT from the Green functions-based calculations. Therefore, a major revision should be applied.

Issue 2.1: Figure 2: “For the Dyphenyl-ether molecule a G = 4 eV was used, and for the Dyphenyl-ether molecule a G = 0.5 eV was used”. Please, correct this misprint, and specify correct Gamma.

Response 2.1: The authors would like to express our most sincere gratitude to the Referee for his/her evaluation of our article. His/her opinions and comments have helped us to substantially improve our work.

The respective changes have been made in the caption of figure 2.

“… bis-(4-mercaptophenyl)-ether molecule a  eV was used, and for the Dyphenyl-ether molecule a eV was used.”

Issue 2.2: Figure 4: Correct the legend, “111” should be “11-1” etc.

Response 2.2: The respective changes have been made to the legends in figure 4.

Issue 2.3: Figure 5: The right upper Y axis should be logarithmic. Please, add a color legend explaining the bottom part of each panel.

Response 2.3: The respective change has been made in the legend of the axes of the upper plots of figure 5, and in the caption of the figure.

“… In the upper part, the red line corresponds to the logarithm of the transmission for a value of  eV (weak coupling), and the blue curve corresponds to the logarithm of the transmission for a value of eV (strong coupling).”

Issue 2.4: Figure 6: Such potentials as +/-6V are high enough for the oxidation processes. The real system will not survive in these conditions.

Response 2.4: We appreciate the referee for their comment. Indeed, when values are within the range of ±6 volts, the molecule starts to exhibit oxidation states. Therefore, it has been decided to adjust the voltage scale from -4 to 4 volts, which is the point of saturation for the molecule.

Issue 2.5: Figure 10: add ZT values at E=EF for all models.

Response 2.5: ZT values have been added in table 1.

Issue 2.6: Calculation of ZT requires absolute values of electrical, thermal conductance, and Seebeck coefficient. However, Figures 7-9 present only relative values. Explain exactly, how the ZT values have been obtained.

Response 2.6: The ZT calculation was performed using the Landauer integrals (second definition of Equation 16), because by replacing the definitions of physical quantities, such as the Seebeck coefficient, electrical conductance, and thermal conductance, the constants vanish, and the ZT is only expressed in terms of Landauer integrals.

Round 2

Reviewer 2 Report

The authors have made all necessary corrections.

Author Response

20th Junio 2023, Medellin

Ms. Max Ma

Prof. Dr. Alexey Kavokin

Prof. Dr. Helgi Sigurdsson

Guest Editors

Physics of Light-Matter Coupling in Nanostructures

Condensed Matter

MDPI

Dear Editors:

Thank you for sending us the report of the manuscript entitled “Theoretical study of thermoelectric properties of a single molecule of Diphenyl-ether” by Rafael G. Toscano-Negrette, José C. León-González, Juan A. Vinasco, Judith Helena Ojeda Silva, Alvaro L. Morales and Carlos A. Duque, with Manuscript ID: condensedmatter-2420546.

In this resubmitted version we have dealt with all the remarks given by the Editor. Our detailed answers to the report, are given below. In the new version of the manuscript the changes are written in red color.

Finally, we would like to thank the Editor for the careful consideration of our submission, and pertinent comments that have helped us to improve the quality and readability of the manuscript. We hope that this resubmitted version is suitable for the publication.

Yours sincerely,
